# User Acceptance of Hotel Service Robots Using the Quantitative Kano Model

**Muzi Xie [1] and Hong-bumm Kim [2,\*]**

1    Graduate School of Hospitality & Tourism Management, Sejong University, Seoul 05006, Korea; xmzmomo@163.com
2    College of Hospitality & Tourism Management, Sejong University, Seoul 05006, Korea
*    Correspondence: kimhb@sejong.ac.kr

**Abstract:** With today's rapid technological developments, many have applied artificial intelligence and robot technology to the tourism and hotel industries, with hotel service robots (HSRs) being gradually developed. At present, more technology development companies have focused their attention on improving HSRs' different attributes to improve their acceptance by users, thereby enhancing market competitiveness and improving customer loyalty. Understanding consumer acceptance of HSRs is important. Based on a literature review of the user's acceptance of HSR attributes and HSRs' current development status, some factors and attributes were extracted. For the questionnaire's design and data extraction, the quantitative Kano model was used. The data obtained were compiled and analyzed using Microsoft Excel and SPSS. This study aims to (1) qualitatively apply the perceived value theory to develop specific HSR attributes and (2) quantitatively examine these attributes concerning public acceptance. By integrating the Kano model with the perceived value theory, this study provides empirical evidence of a nonlinear relationship between HSRs' perceived value and user acceptance by exploring various attributes affecting the user's acceptance of HSRs and the degree of change brought by the different attributes. The research result reveals the multidimensional impacts of perceived value, prompting users to embrace newer HSR technologies.

**Keywords:** hotel service robots; Kano model; user acceptance

## 1. Introduction

Service robots are semi- or fully-autonomous robots that perform services contributing to the well-being of humans and equipment. They do not include manufacturing operations, and they make decisions and perform autonomous actions in a real, unpredictable environment [1]. Hotel service robots (HSRs) bring many societal benefits. The rapid development of robotics, automation, and AI effects and changes many aspects of the hotel and service industries [2]. For example, Japan's Henn-na Hotel, which opened in 2015, was the first hotel operated by robots, according to the Guinness Book of World Records. It uses anthropomorphic robots to replace the services of human employees [3]. From front desk check-in to automatic baggage delivery and guest room companionship, each step uses HSRs [4]. Another example is Hilton Worldwide Hotels, in cooperation with IBM, which trialed the world's first robotic concierge service, which mainly introduced guests to local attractions, restaurants, and facilities [4]. Hotel Jen and Hotel Icon, in Singapore and Hong Kong, respectively, also use different HSRs to replace or assist human workers and reduce labor costs [3].

While HSRs have many advantages, they also face challenges. For example, the price is higher (increase in the room occupancy rate), or there are poor experiences that cause consumer dissatisfaction, such as complex applications and difficulty in studying and usage, self-check-in, and face recognition [5]. Moreover, the smooth communication between users and AI is also a problem. AI's HSR is a new technology, so it faces security and personal privacy issues [6]. Some studies believe using HSRs can change the users'

evaluation and acceptance of hotels [7]. Theoretical studies have determined and examined the public acceptance of HSR's determinants. for example, the device's appearance or its movement characteristics [8]. There are also perceived safety and intelligence issues that affect the use of HSRs [6].

However, there are a few limitations within the current literature. First, existing studies have not analyzed factors that influence public acceptance from a value creation perspective. In marketing terminology, perceived value is an individual evaluation of a product or service's merits and ability to meet individual needs and expectations. Value is a measurement of the benefits provided by goods or services to an economic agent. Understanding HSRs' value will allow more people to accept HSRs. Perceived value is a multidimensional structure, including various concepts such as perceived price, quality, benefit, and sacrifice [9]. This study proposes to enhance the public's perceived value in accepting HSRs because of their economic, emotional, quality/function, and social/environment abilities. Consequently, the public's perceived value enhances their acceptance of HSRs. Second, existing studies assumed linear relationships between determinants and HSRs' public acceptance. Certain criteria do not consider the general public's actual needs, with continuous improvement not sufficient enough to increase public acceptance. Traditional methods may not fully explain the relationship between HSRs' general acceptance and their perceived value. Furthermore, two-dimensional (linear and non-linear) quality models can solve linear quality models' shortcomings [10].

The purpose of this study is to develop specific HSR attributes through the qualitative application of perceived value theory, together with a quantitative examination of the attributes related to public acceptance. To explicate the perceived values' compositions and differing impacts on public acceptance, a decomposed approach is adopted in this study, particularly based on synthesized insights of the Kano model [11], By integrating the Kano model with the perceived value theory, this study contributes by providing theoretical and practical insights regarding not only what the perceived value attributes are but also how they collectively contribute to public acceptance.

## 2. Literature Review

### 2.1. Contemporary Research on Public Acceptance of HSRs

The rise of robotics has spanned many fields and experienced decades of processes before finally entering the service industry. Companies are designing more robots to function as part of ordinary people's lives. Increasingly advanced artificial intelligence (AI), robotics, and machine learning technologies enable providers to provide higher speed and efficiency, with the ability to replace most of the labor force [12]. Many of the robot's tasks range from entertainment to assisting humans in completing difficult or tedious tasks and even socializing and communicating with and serving humans [13].

Service robots and AI promise to increase productivity, reduce costs, drive sales, and promote a substantial increase in applied research [14]. According to a survey of business leaders, 24% of US companies already use AI, with 60% wanting to use AI in 2022 [12]. With sales of service robots continuing to grow, the next ten years will see the further expansion of using personal and professional service robots. With its rapid development, more scholars are studying service robots [13]. However, the existing research focuses on the providers' perspectives and on the robot's design and performance [4], for example, image recognition and their appearance. Some researchers have also studied their impact on hotel operations (to reduce costs) [15], with others study their impact on company management (the possibility of robots and employees working together to optimize work efficiency) [15]. Many kinds of research are conceptual and descriptive [16].

Although some studies initially explored the attributes of user acceptance, most were based on the traditional technological acceptance theories [4,17]. The Technology Acceptance Model posits that two beliefs, perceived usefulness and ease of use, determine an intention to use technology. To understand the user's acceptance of HSR attributes more comprehensively, one must analyze each attribute from a multidimensional perspective,

which is helpful to HSRs' development. Therefore, this study adopts the perceptual value theory to decompose the acceptance of audiovisual public development. Kano and Mulavwa [10] developed linear and non-linear two-dimensional quality models to solve the conventional linear quality model's shortcomings, which may not fully explain the relationship between the public's acceptance of HSRs and the perceived value.

*2.2. Decomposing Public Development Acceptance of HSRs: Application of the Perceived Value Theory*

The perceived value theory posits that the customers' perceived value derives from evaluating the product or service's benefits and costs. The concept of perceived value implies the interaction between the consumer and the product; unidimensional and multidimensional models of a value play a role in providing a unidimensional and multidimensional understanding of the concept; its nature is multidimensional and complex, and that value is relative because it is comparative, personal and situational [9]. This study proposes to enhance the perceived value of the public accepting HSRs by providing information on their economic, emotional, quality/function, and social/environment abilities. The users' acceptance of HSRs would be different according to how they perceive the various attributes of HSRs.

Economic factor (EF): HSRs' economics means their use can bring economic or property benefits to users in scenarios such as hotel rooms, services, and catering prices. Studies have shown that the amount spent in hotels affects users' acceptance and future choices [18]. When hotel users experience specials, offers, and discounts on hotel rooms, food, and other services, they experience fun, pleasure, and enjoyment, and the price has a significant positive impact on user satisfaction [19]. If the number of HSRs increases, the hotel room, service, and catering prices change. For example, when adding HSR projects, hotel service prices will be reduced, giving users positive benefits. Contrarily, when adding HSR items, the room price will rise, giving users negative benefits [20]. Therefore, using HSRs can bring economic or property benefits in some aspects, although users may have different attitudes and opinions.

- Hotel room price: The amount paid for staying in a hotel room. (A1)
- Hotel charge service price: The price when using dry cleaning services. (A2)
- Catering price: The cost incurred when purchasing food or drinks. (A3)

Quality/function (QF): Quality is a somewhat subjective, conditional, and perceptual attribute. Its understanding varies from person to person. Users may measure quality to the degree that an HSR is safe, reliable, and time-saving, has personification, is convenient, diversified, clean, and quiet, and has privacy protection. For example, when staying in a smart hotel, the leakage of user privacy can cause users to be reluctant to share their data [21]; the anthropomorphic appearance or behavior of HSRs affects user acceptance to varying degrees [22]; the safety of HSRs has an impact on customers [23]; customers are more inclined to use the time-saving services [24]; with diversified services being more accepted, the hotel is more attractive [25].

- Safety: The safety of using robots. (A4)
- Reliability: Robots can accurately execute commands. (A5)
- Time-saving: It takes less time for the robot to execute commands. (A6)
- Personification: It feels more like a real human being. (A7)
- Convenience: Simpler and faster. (A8)
- Diversified Services: More types and forms of services. (A9)
- Clean: The cleanliness of the hotel's environment. (A10)
- Quiet: The surrounding environment is pleasant during their stay. (A11)
- Privacy protection: The high-tech means of robots make the personal privacy of customers better protected. (A12)

Emotional (EM): Emotion refers to the positive feelings users have when using HSRs. Users have many subjective emotions when experiencing services, with their emotions also affecting their HSR acceptance. This study's emotions include trustworthiness, luxurious-

ness, social interaction, appearance, and enjoyment. For example, studies show that in the hotel service environment, customers will evaluate services based on whether they are pleasant and entertaining [26]; the appearance of the hotel inside and outside is one of the important factors for the successful development of the hotel [27]; the social interaction of customers in the hotel affects the quality of customer experience [28]. Pan, et al. [29] believed that HSRs' appearance also affects user acceptance.

- Trustworthiness: In an uncertain environment, the user actively predicts the robot's behaviors, believing it will act as expected. (A13)
- Luxuriousness: A magnificent, rich feeling. (A14)
- Social interaction: Social activities interacting with other individuals for material and spiritual exchanges. (A15)
- Appearance: Looks comfortable and happy. (A16)
- Enjoyment: The feeling of pleasure and satisfaction when one does or experiences something positive. (A17)

Social/environment (SE): Social/environment refers to the positive impact HSR applications have on society and the environment. The public must recognize HSRs for them to develop rapidly. Social/environment includes legal and social norm compliances, reduced manufacturing waste, reputation, and scientific studies. For example, using robots reduces the output of waste and effectively coordinates waste management [30]. In the development and evolution of robots, the ethical issues and legal ethics of robots in certain environments have also received great attention [31]. In terms of customer loyalty, hotel reputation plays an important role [32].

- Legal compliance: Compliance with the law. (A18)
- Social norm compliance: Robots make more ethical choices. For example, if a guest falls or has a sudden illness, the robot automatically calls the police or helps the guest. (A19)
- Reduced manufacturing waste: Precise use of computing resources without waste. (A20)
- Reputation: Famous for using the latest technology of hotel robots. (A21)
- Scientific: Using hotel robots promotes science's necessity and enables people to support using and developing new technologies. (A22)

*2.3. Differentiating Attributes of Public Perceived Value: Application of the Kano Model*

First, to understand the user's acceptance of the HSR's nature, the 20 decomposed product attributes exert different impacts on public acceptance. The Kano model is a useful tool to assist product development and user satisfaction assessment. It does not use simple linear relationships; instead, it identifies the different effects of hotel robot attributes. Some attributes cause users' unexpected satisfaction or overwhelming disgust, with their acceptance affected to varying degrees [33]. The Kano model can help hotel service robot developers better understand user needs and how these needs affect user acceptance and then focus on the tasks that most affect user acceptance and satisfaction in order to remain competitive and gain greater profits. For those involved in the hotel and tourism industry, this has an important impact on the design of optimized resource allocation schemes. The Kano model divides product attributes into five categories [34]:

Attractive (A): This involves positive emotions, such as happiness and surprise, and, when satisfied, causes users to accept them. When these attributes are highly provided, users have a larger acceptance percentage. However, when the attributes are lower, users will not have a large amount of disapproval. The attributes' high performance leads to over-proportional acceptance, while lower performances do not cause users to not accept them. These attributes are the product standards that impact customer satisfaction the most with a given product. The client neither clearly expresses nor expects attractive requirements. Satisfying these attributes leads to proportional satisfaction. However, if these attributes are not satisfied, there will be no feeling of disapproval.

One-dimensional (O): These are standard service elements users generally expect. These attributes enhance acceptance when the performance level increases. Regarding these attributes, customer satisfaction is directly proportional to satisfaction—the higher the satisfaction, the higher the customer satisfaction, and vice versa. Customers usually explicitly request one-dimensional attributes.

Must-be (M): This is the basic standard users require. When these attributes are provided at a higher level, users will not bring much acceptance, but when they are provided at a lower level, users will be extremely unaccepting. While customers take these attributes for granted and satisfying them will not increase customer satisfaction, if these attributes are not satisfied, users will feel dissatisfied. Must-be attributes are the product's basic standard. Meeting these prerequisites leads to an "unsatisfied" state. The user takes the necessary attributes as something they should have for granted, with customers regarding them as preconditions, so they are not explicitly required. However, if these attributes are not met, users will lose interest in the product. Therefore, in any case, attributes that must be possessed are the decisive competitive factors.

Indifferent (I): These are insignificant service elements for users. Regardless of whether hotels provide it, it has no impact on the user's experience. They are neither positive nor negative aspects of quality. They do not lead to users being accepting or unaccepting. Investing in these attributes will not increase user acceptance, so these should be avoided.

Reverse (R): This leads to user disapproval when met and user acceptance when not met. It refers to quality characteristics that cause strong dissatisfaction and quality characteristics that lead to lower levels of satisfaction, as not all consumers have similar preferences. Many users do not have this requirement, with user satisfaction declining after providing it. The degree of provision is inversely proportional to the degree of user satisfaction.

Through the Kano model, researchers can more accurately determine the product design and development's key points. For example, the must-be attributes, where investment improvement has reached a satisfactory level, are not useful. Therefore, it is best to improve one-dimensional or attractive attributes as they greatly affect the perceived product quality, which has a greater impact on user acceptance.

After identifying customer needs, levels, and priorities, deploying Kano models and quality features is the best combination [35]. The Kano model is used to determine the importance of individual product characteristics to user satisfaction; hence, it creates optimal prerequisites for process-oriented product development activities that facilitate research trade-offs during the product development phase. If two product requirements cannot be simultaneously met due to economic or technical reasons, then the direction with the greatest user acceptance impact is determined [34]. The Kano model can be used as a prioritization tool [36]. Must-be, one-dimensional, and attractive requirements often differ in the expectations of different customer groups. From this aspect, user-tailored solutions can be developed for special problems, thus guaranteeing the best satisfaction of different user groups. By integrating the Kano model into the decomposed perceived value framework, this study categorizes the proposed product attributes and examines their distinct impacts on users' evaluation of perceived value.

## 3. Methods

### 3.1. Questionnaire Design and Sample Statistics

First, according to the National Bureau of Statistics of China, Beijing, Shanghai, and Guangzhou are China's first three cities to move towards internationalization. The survey was randomly distributed to the public, regardless of age and gender. The questionnaire, published online (wjx.cn accessed on 18 October 2020), randomly asked passers-by in Beijing, Shanghai, and Guangzhou to use mobile phones to scan the QR code to fill it in. First, high traffic locations in three cities were identified: Wangfujing, Tiananmen, and Tiantan in Beijing, Nanjing Pedestrian Street in Shanghai, and the Bund, Oriental Pearl, Guangdong Provincial Bus Station, Tianhe Sports Center, and Tianhe Port in Guangzhou.

A total of 308 questionnaires was collected over a period of 7 days. Statistical examinations deleted 47 invalid questionnaires, with the remaining 261 used as statistical data for further analysis. The interviewees' demographic characteristics are listed in Table 1.

**Table 1.** Descriptive statistical analysis.

| Demographic Characteristics | Number of Respondents (n = 261) | Percentage (%) |
|---|---|---|
| Gender | | |
| Male | 91 | 34.87 |
| Female | 170 | 65.13 |
| Age | | |
| Less than 18 years old | 2 | 0.77 |
| 18–28 years old | 142 | 54.41 |
| 29–38 years old | 72 | 27.59 |
| 39–48 years old | 22 | 8.43 |
| 49–58 years old | 21 | 8.05 |
| More than 58 years old | 2 | 0.77 |
| Educational Status | | |
| High school degree | 25 | 9.58 |
| College degree | 146 | 55.94 |
| Master's degree | 53 | 20.31 |
| Doctoral degree | 14 | 5.36 |
| Not in the option | 23 | 8.81 |
| Profession | | |
| Full-time student | 55 | 21.07 |
| Production staff | 10 | 3.83 |
| Salesperson | 19 | 7.28 |
| Financial auditor | 17 | 6.51 |
| Civilian staff | 15 | 5.75 |
| Teacher | 42 | 17.5 |
| Consultant | 2 | 16.09 |
| Firm employees | 59 | 22.61 |
| Professionals | 8 | 3.07 |
| Other (self-employed, freelance, farmer, unemployed, transport, etc.) | 34 | 13.02 |
| Annual income level | | |
| 50,000–100,000 | 119 | 45.59 |
| 100,000–200,000 | 50 | 19.16 |
| 200,000–300,000 | 13 | 4.98 |
| Higher than 300,000 | 12 | 4.6 |
| Not in the option | 67 | 25.67 |

According to the respondents' genders, 91 (34.87%) were male and 170 (63.13%) were female. In the survey, there were 14 people (5.36%) with PhDs, 53 (20.31%) with a master's degree, 146 (55.94%) with a bachelor's degree, 25 (9.58%) with high school education, and

23 (8.81%) not in the above options. Regarding their profession, the number of full-time students was the highest (55 (21.07%)), followed by teaching 42 (16.09%), and 34 (13.03%) with other professions (such as self-employed, freelancer, and farmer). Of the respondents' total annual income, in RMB (yuan), 80 (45.59%) were between 50,000 and 10,000 yuan, the largest proportion of all respondents. This was followed 100,000 to 200,000 (50, 19.16%), 200,000 to 300,000 (13, 4.98%), and 12 people (4.6%) with an annual income of more than 300,000 yuan. In addition, 67 (25.67%) are not in the options.

Sample statistics show that the research sample group comprises users from different regions, genders, and ages. Respondents have varying types of work, and their annual income levels and academic qualifications vary. Their age ranges from less than 18 years old (54.41%) to over 58 years old (0.77%). From the perspective of income structure, only 4.6% of users had an annual income of more than 300,000, with more than 50% of users having an annual income of less than 100,000. The annual income may affect consumption concepts. Users have differing price preferences and concepts of hotel consumption.

This study employs the basic Kano method, proposed in 1984. The questionnaire for this study in Appendix A followed the Kano model design questions. The Kano method includes the inverse question of paired functional and dysfunction problems and uses a structured questionnaire to determine the acceptance of HSR's attributes. The functional problem captures the user's feelings when certain attributes are met, while dysfunction problems capture feelings when the attributes are not met. Through the questionnaire, must-be (M), one-dimensional (O), and attractive (A) attributes are classified, as well as the product attributes the users are indifferent (I) to. For example, for the price attribute, functional and dysfunctional questions are, "If hotel room prices for hotel service robots are not cheap, would you choose it?" and "If hotel room prices for hotel service robots are cheap, would you choose it?" For each question, the user is asked to choose from five answers: "① I like it that way, ② It must be that way, ③ I am neutral, ④ I can live with it that way, ⑤ I dislike it that way".

It then uses the classification table (Table 2) to analyze the user's answers [37]. For example, for a certain HSR attribute, if the user's answer to a functional question is "I like that" and the answer to a dysfunction question is "I can do that", the attribute is classified as A (attractive). If satisfying this product attribute brings a pleasant feeling to the user, the user chooses "I like that", and the dissatisfaction does not necessarily lead to unacceptance, with the user choosing "I can live with it that way". If the user chooses "I like it that way" and "I dislike it that way" to answer functional and dysfunctional questions, respectively, this attribute will belong to the O (one-dimensional) category. This shows that the product attribute creates proportional user acceptance (unacceptance) when fulfilled (unfulfilled).

**Table 2.** Classification table of the Kano method.

| User Requirement | | Answer to the Dysfunctional Question | | | | |
|---|---|---|---|---|---|---|
| | | Like | Must Be | Neutral | Live with | Dislike |
| Answer to the functional question | Like | Q | A | A | A | O |
| | Must be | R | I | I | I | M |
| | Neutral | R | I | I | I | M |
| | Live with | R | I | I | I | M |
| | Dislike | R | R | R | R | Q |

A: attractive; O: one-dimensional; M: must-be; I: indifferent; R: reverse; Q: questionable.

If the answers combine to arrive at category I, it means the user is indifferent to the product function. The user does not care if it exists. Hence, users are unwilling to spend more money on this feature. Class Q represents suspicious results. Usually, the answer does not fall into this category and is a suspicious score, indicating incorrect wording of the question or that the respondent misunderstood it or mistakenly crossed out the wrong answer. If looking up the answer in the evaluation form results in category R, then

customers not only do not want to use this product feature but even hope to get the opposite result. It then classifies and summarizes the answers based on each user to determine the last attribute category [38]. Besides the above five categories, some illogical reactions are introduced as questionable (Q). It is rare for product attributes to be classified as Q (questionable) and R (reverse). Therefore, the focus is on the I (indifferent), A (attractive), M (must), and O (one-dimensional) categories.

### 3.2. Quantitative Acceptance Analysis

User acceptance coefficients reveal much information, such as whether it can improve user acceptance through meeting product attribute requirements or whether it prevents users from not accepting products by meeting product attribute requirements. In different markets, there are different needs and requirements, so one can refer to it when one does not know which category the product function is allocated. One must understand the average impact of product demand on all user acceptance. The user acceptance coefficient indicates the degree to which the product function affects user acceptance or to which users do not accept it if the product function is not satisfied [39].

The average effect on acceptance is calculated by adding the attractive and one-dimensional columns before dividing the summed total of attractive, one-dimensional, must-be, and indifferent columns. The average effect on unacceptance is calculated by adding the must-be and one-dimensional columns and dividing by the same normalization factor [34]. It is important for developers and consumers to know which properties to focus on. This method uses the following formulas to calculate better and worse values to see how satisfied and dissatisfied the user is with the attribute to judge user acceptance [40].

The extent of acceptance:

$$\frac{A+O}{A+O+M+I}$$

The extent of unacceptance:

$$\frac{O+M}{(A+O+M+I)\times(-1)}$$

Adding a minus sign in front of the user acceptance coefficient means the user does not accept the attribute, emphasizing that the product attribute does not accept the negative impact on the user acceptance.

In this study, user acceptance analysis is conducted by extending the Kano method from a quantitative perspective. The user acceptance performance (U-P) was calculated using the satisfactory performance (S-P) function proposed by Wang and Ji [41]. The user's like ($d^+$) and dislike ($d^-$) levels quantify each attribute of the HSR, as follows:

$$d^+ = \frac{P_i^A + P_i^O}{P_i^A + P_i^O + P_i^M + P_i^{I\prime}} \tag{1}$$

$$d^- = \frac{P_i^O + P_i^M}{P_i^A + P_i^O + P_i^M + P_i^{I\prime}} \tag{2}$$

The corresponding percentages of the Carnot category for a given attribute *i* are denoted as $P_i^A$(A), $P_i^O$(O), $P_i^M$(M), and $P_i^I$(I). Here, the $d^+$ value reflects the user acceptance's impact when product requirements are met, with the $d^-$ value reflecting the impact when product requirements are not met. Therefore, taking attribute A as an example in Figure 1, the degree of user acceptance ($U_i$) for a given attribute i is expressed by interpolating the two endpoints of $(1, d_i^+)$ and $(0, d_i^-)$. Thus, $U_i$ is thought of as a weighted average of user delight ($d^+$) and disgust ($d^-$) at a given performance level of 0 to 1.

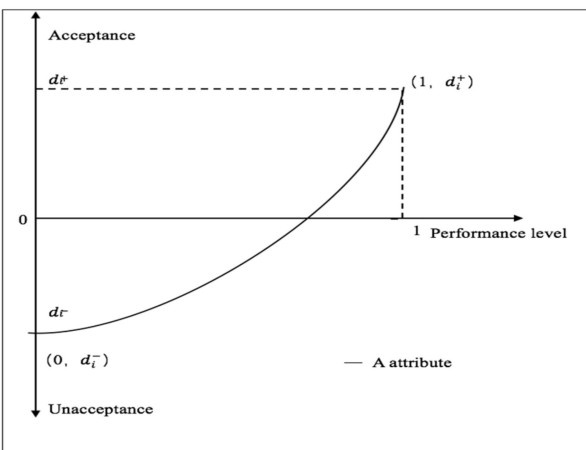

**Figure 1.** Typical relationship curve of (un)acceptance and product performance level for the attractive (A) attribute.

For users regarding attributes such as O, the linear relationship between the performance level and user acceptance for a specific attribute i is expressed as $U_i^O = a_i x_i + b_i$; the performance level of attribute i is expressed as $x_i$, adjusting the Kano curve of the parameters for slope and intercepting, denoted by $a_i$ and $b_i$. Substituting the two endpoints of $(0, d_i^-)$ and $(1, d_i^+)$ into the equation gives $a_i = d_i^+ - d_i^-$ and $b_i = d_i^-$. Therefore, the U-P function for O attributes is expressed as follows:

$$U_i^O = d_i^+ - d_i^- x_i + d_i^-. \tag{3}$$

For users viewing certain attributes as A, the U-P function is estimated using an exponential function $U_i^A = a_i e^{x_i} + b_i$. Similar to O attributes, substituting two endpoints of $(1, d_i^+)$ and $(0, d_i^-)$ into the equation gives $a_i = \frac{d_i^+ - d_i^-}{e-1}$ and $b_i = -\frac{d_i^+ - ed_i^-}{e-1}$. Therefore, the U-P function for A attributes is expressed as follows:

$$U_i^A = \frac{d_i^+ - d_i^-}{e - 1} e^{x_i} - \frac{d_i^+ - ed_i^-}{e - 1}. \tag{4}$$

For users viewing certain attributes as M, the U-P function is expressed by an exponential function $U_i^M = a_i (-e^{-x_i}) + b_i$, where $a_i = \frac{e(d_i^+ - d_i^-)}{e-1}$ and $b_i = \frac{ed_i^+ - d_i^-}{e-1}$ are substituted for the two endpoints. Hence, the U-P function for M attributes is as follows:

$$U_i^M = \frac{e(d_i^+ - d_i^-)}{e - 1} e^{-x_i} + \frac{ed_i^+ - d_i^-}{e - 1}. \tag{5}$$

Therefore, the overall group acceptance of a given product attribute *i* is calculated using the following function:

$$U_i = U_i^O P_i^O + U_i^A P_i^A + U_i^M P_i^M + U_i^I P_i^I, \tag{6}$$

where $U_i^I$ is the acceptance level derived from the I product attribute i, which equals 0.

The user acceptance positive factor ranges from 0 to 1. The closer the value is to 1, the greater the impact on user acceptance. However, the negative user acceptance factor must be considered. If it is close to −1, the impact on user unacceptance is great if the analyzed product features are not met. A value close to 0 indicates that this feature will not cause user rejection. The value of each attribute can indicate the product attribute's importance in the competition. From the user's perspective, the higher the value is in the positive range, the higher the relative competitive advantage of perceived product attributes. Conversely, the higher the index's value is in the negative range, the greater the relative competitive

disadvantage. Therefore, it is necessary to focus attention on improving the requirements for product attributes.

## 4. Results

### 4.1. Quantitative Acceptance Analysis

A qualitative classification of HSR attributes may not adequately interpret user needs. This study employed a quantitative analysis of the Kano model to establish more comprehensive quantitative relationships between product performance levels and user (un)acceptance. First, to calculate values of user delight ($d^+$) and disgust ($d^-$) for each product attribute, it uses Equations (1) and (2) (see Table 3, columns 2 and 3). Next, by substituting $d^+$ and $d^-$ into Equations (3)–(5), Columns 4–6 of Table 3 are the user's (un)acceptance of product attributes perceived as ($U_O$), A ($U_A$), and M ($U_M$) as a function of $x_i$ (performance level). Lastly, the (un)acceptance of a given product attribute i ($U_i$) by the entire user group is calculated by Formula (6) as the weighted average of $U_O$, $U_A$, and $U_M$, as shown in Table 3, column 7.

**Table 3.** Quantitative acceptance analysis.

| (1) | (2) | (3) | (4) | (5) | (6) | (7) |
|---|---|---|---|---|---|---|
| | $d^+$ | $d^-$ | $U_i^O$ | $U_i^A$ | $U_i^M$ | $U_i = U_i^O P_i^O + U_i^A P_i^A + U_i^M P_i^M + U_i^I P_i^I,$ |
| A1 | 0.5787 | −0.3701 | $0.95x_i - 0.37$ | $0.55x_i - 0.92$ | $-1.50x_i + 1.13$ | $0.21x_i + 0.19e^{x_i} - 0.21e^{-x_i} - 0.24$ |
| A2 | 0.5099 | −0.3992 | $0.91x_i - 0.40$ | $0.53x_i - 0.93$ | $-1.44x_i + 1.04$ | $0.22x_i + 0.13e^{x_i} - 0.20e^{-x_i} - 0.18$ |
| A3 | 0.5320 | −0.3880 | $0.92x_i - 0.39$ | $0.54x_i - 0.92$ | $-1.46x_i + 1.07$ | $0.22x_i + 0.14e^{x_i} - 0.19e^{-x_i} - 0.20$ |
| A4 | 0.5219 | −0.6693 | $1.19x_i - 0.67$ | $0.69x_i - 1.36$ | $-1.88x_i + 1.22$ | $0.50x_i + 0.06e^{x_i} - 0.43e^{-x_i} - 0.12$ |
| A5 | 0.5984 | −0.6299 | $1.23x_i - 0.63$ | $0.71x_i - 1.34$ | $-1.94x_i + 1.31$ | $0.55x_i + 0.10e^{x_i} - 0.32e^{-x_i} - 0.25$ |
| A6 | 0.5814 | −0.4651 | $1.05x_i - 0.47$ | $0.61x_i - 1.07$ | $-1.66x_i + 1.19$ | $0.34x_i + 0.15e^{x_i} - 0.22e^{-x_i} - 0.26$ |
| A7 | 0.5664 | −0.6055 | $1.17x_i - 0.61$ | $0.68x_i - 1.29$ | $-1.85x_i + 1.25$ | $0.52x_i + 0.08e^{x_i} - 0.28e^{-x_i} - 0.22$ |
| A8 | 0.5922 | −0.5608 | $1.15x_i - 0.56$ | $0.67x_i - 1.23$ | $-1.82x_i + 1.26$ | $0.47x_i + 0.12e^{x_i} - 0.26e^{-x_i} - 0.26$ |
| A9 | 0.5422 | −0.4337 | $0.98x_i - 0.43$ | $0.57x_i - 1.00$ | $-1.54x_i + 1.11$ | $0.26x_i + 0.14e^{x_i} - 0.23e^{-x_i} - 0.20$ |
| A10 | 0.6235 | −0.6784 | $1.30x_i - 0.68$ | $0.76x_i - 1.44$ | $-2.06x_i + 1.38$ | $0.65x_i + 0.08e^{x_i} - 0.33e^{-x_i} - 0.27$ |
| A11 | 0.6310 | −0.6548 | $1.29x_i - 0.65$ | $0.75x_i - 1.40$ | $-2.03x_i + 1.38$ | $0.61x_i + 0.10e^{x_i} - 0.33e^{-x_i} - 0.28$ |
| A12 | 0.6142 | −0.6890 | $1.30x_i - 0.69$ | $0.76x_i - 1.45$ | $-2.06x_i + 1.37$ | $0.66x_i + 0.07e^{x_i} - 0.34e^{-x_i} - 0.26$ |
| A13 | 0.6048 | −0.6371 | $1.24x_i - 0.64$ | $0.72x_i - 1.36$ | $-1.96x_i + 1.33$ | $0.58x_i + 0.08e^{x_i} - 0.28e^{-x_i} - 0.26$ |
| A14 | 0.5238 | −0.3294 | $0.85x_i - 0.33$ | $0.50x_i - 0.83$ | $-1.35x_i + 1.02$ | $0.19x_i + 0.14e^{x_i} - 0.12e^{-x_i} - 0.21$ |
| A15 | 0.6653 | −0.3745 | $1.04x_i - 0.37$ | $0.61x_i - 0.98$ | $-1.65x_i + 1.27$ | $0.31x_i + 0.21e^{x_i} - 0.10e^{-x_i} - 0.37$ |
| A16 | 0.5178 | −0.3439 | $0.86x_i - 0.34$ | $0.50x_i - 0.85$ | $-1.36x_i + 1.02$ | $0.17x_i + 0.15e^{x_i} - 0.18e^{-x_i} - 0.19$ |
| A17 | 0.6335 | −0.4382 | $1.07x_i - 0.44$ | $0.62x_i - 1.06$ | $-1.70x_i + 1.26$ | $0.34x_i + 0.18e^{x_i} - 0.18e^{-x_i} - 0.32$ |
| A18 | 0.5630 | −0.6693 | $1.23x_i - 0.67$ | $0.72x_i - 1.39$ | $-1.95x_i - 1.28$ | $0.57x_i + 0.06e^{x_i} - 0.37e^{-x_i} - 0.19$ |
| A19 | 0.6142 | −0.6024 | $1.22x_i - 0.60$ | $0.71x_i - 1.31$ | $-1.92x_i + 1.32$ | $0.56x_i + 0.10e^{x_i} - 0.24e^{-x_i} - 0.29$ |
| A20 | 0.6200 | −0.4880 | $1.11x_i - 0.49$ | $0.64x_i - 1.13$ | $-1.75x_i + 1.26$ | $0.39x_i + 0.16e^{x_i} - 0.20e^{-x_i} - 0.30$ |
| A21 | 0.5652 | −0.3478 | $0.91x_i - 0.49$ | $0.53x_i - 0.88$ | $-1.44x_i + 1.10$ | $0.25x_i + 0.14e^{x_i} - 0.09e^{-x_i} - 0.27$ |
| A22 | 0.5866 | −0.4843 | $1.07x_i - 0.48$ | $0.62x_i - 1.11$ | $-1.69x_i + 1.21$ | $0.38x_i + 0.14e^{x_i} - 0.20e^{-x_i} - 0.27$ |

This study selectively draws a scatter plot *i* to observe the relationship between U and *x*, where the *x*-axis shows the user's unacceptance at the 10% lower product performance level, and the *y*-axis shows user acceptance at 90% higher product performance. In Figure 2, the product attribute plotted in the shaded area in the upper right corner of the chart is similar to the A product attribute in the Kano model. These properties contribute more to

acceptance formation when performed at relatively higher levels than when performed at lower levels. The attributes within the shaded area in the lower-left corner are similar to the M product attributes in the Kano model, which results in more disapproval than acceptance. Lastly, product attributes along the dashed diagonal line lead to proportional (un)acceptance when (not) fulfilled.

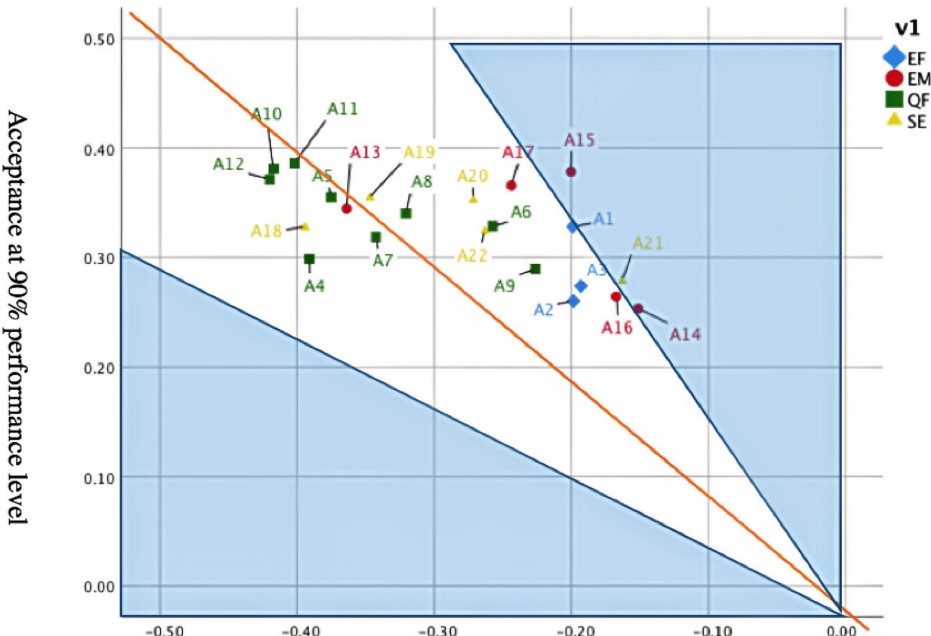

**Figure 2.** Acceptance versus unacceptance.

For instance, looking at coordinate A4, a lower (10%) level of product performance on the *x*-axis resulted in an unacceptance of around $-0.39$, but a higher (90%) level of product performance on the *y*-axis caused only an acceptance of around 0.29. Since improving the performance level of the product greatly reduces the users' unacceptability, A4 is more worthy of investment when the product performance level is lower, but A4 is not worth further investment when the performance level is higher, as further improving the performance level will not lead to greater user acceptance.

However, the A15 in the upper right shaded area causes only around $-0.20$ unacceptance when its performance level is lower, but more acceptance (around 0.38) when its performance level is higher. Hence, A15 is the opposite of A4, with A15 being a more valuable investment when the product performance level is higher and being ignored if the product performance level is lower. From this, it appears that attributes closer to the diagonal line contribute more evenly to user acceptance and unacceptance.

Attributes can also be compared within groups from the same dimension in Figure 2. In EF, A1 (Hotel room price), A2 (Hotel service charge price), and A3 (Catering price) all contribute more to acceptance when the product performance level is higher (90%), with A1 having the largest contribution value; the hotel room price is the more worth investment to gain greater acceptance. In EM, A13 (Trustworthy) has a larger contribution to (un)acceptance at both higher (90%) and lower (10%) product performance levels, while A14 (Luxury) has a relatively small contribution. Hence, in EM, trustworthy investments should be made to obtain greater benefits. A10 (Clean), A11 (Quiet), and A12 (Privacy protection) in QF are also the same as A13 in EM, with their contribution to (un)acceptance being relatively large, while A9 (Diversified Services) is relatively small. In SE, A18 (Legal Compliance) contributes more to unacceptance at a lower (10%) product performance level, so increasing product legal compliance at an early stage of product development can reduce user unacceptance.

## 4.2. Sensitivity Analysis: Acceptance Formation Rate versus Product Performance

Figure 3 shows the results of computing the first derivative of $U_i$. ($U'_i$.) to test the sensitivity of user (un)acceptance to product performance. Figure 3 shows the results. Among the 22 service attributes, user A15 (Social interaction) has the strongest response: the better the social interaction, the higher the user acceptance. Additionally, its impact on user acceptance increases as performance levels continue to increase. Therefore, A15 (Social interaction) should be first among the 22 attributes for consideration. In Figure 3, A1 (Hotel room price), A10 (Clean), A11 (Quiet), A12 (Privacy protection), A5 (Reliability), A8 (Convenience), A13 (Trustworthy), A15 (Social interaction), A17 (Enjoyment), A19 (Social norms compliance), and A20 (Reduced manufacturing waste) are all above average; hence, from a resource allocation perspective, these attributes may be prioritized.

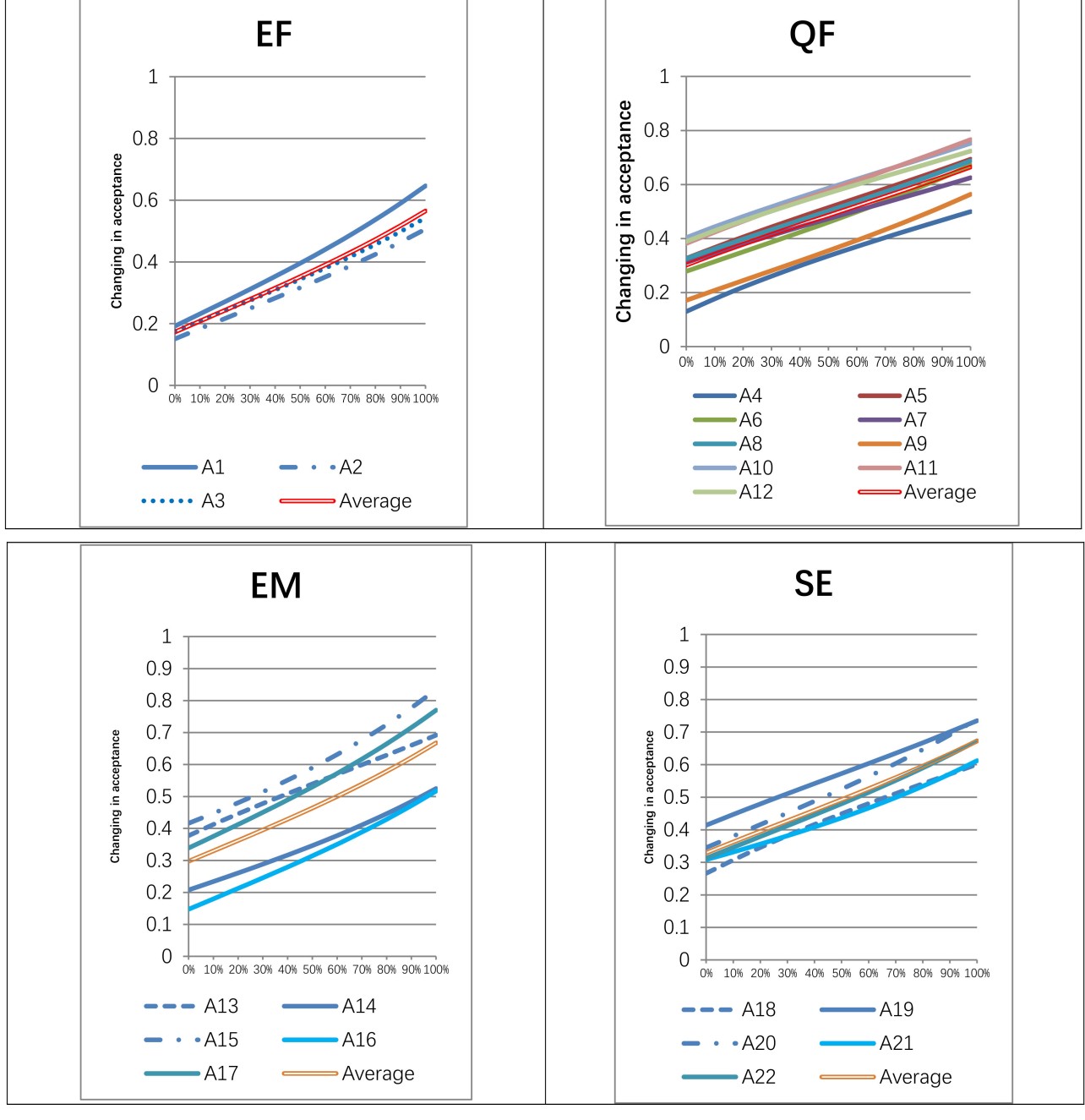

**Figure 3.** Sensitivity test of individual product attributes.

In EF (Social/environment), A1 (Hotel room price) has the highest inclination among these three product attributes, with users having the strongest response to A1: the lower the hotel room price, the more effective it is to form user acceptance. Additionally, as performance levels continue to increase, its impact on user acceptance increases. Hence, in EF (Social/environment), the hotel room price is the first consideration. In QF (Quality/function), as shown in Figure 3, the tendencies of A10 (Clean), A11 (Quiet), and A12 (Privacy protection) are all high in QF, and the values are close together, but, with A12, when the service performance is lower than at 30%, the response of A12 is not as strong as A10. When the service performance is higher than 30%, the response of A12 is not as strong as A11. Hence, in QF, A10 and A11 are the first factors for consideration. The higher the level of the Clean and Quiet attributes, the more effective it is to form user acceptance. In EM (Emotional), A15 (Social interaction) has the strongest response, so from the perspective of EM, A15 is the first factor for consideration: the more social interaction improves, the more user acceptance will form. A13 (Trustworthy) and A17 (Enjoyment), which are both above the horizontal value, have the same degree of response when the service performances are 60%; the two lines intersect. When the service performance is less than 60%, the A13 attribute is more responsive, and when the service performance exceeds 60%, A17 is more responsive.

Therefore, when the service performance reaches a certain level, it is necessary to reallocate resources to avoid a reduction in revenue. In SE (Social/environment), A19 (Social norms compliance) is the first factor for consideration: the more the social norms compliance attribute improves, the more effective it can be to form user acceptance. A20 (Reduced manufacturing waste) rapidly approaches A19 in response to when service performance increases. When service performances are close to 100%, A20's response exceeds A19's, indicating A20 is also a potential attribute worth considering.

Lastly, by assuming that users' total acceptance of HSR attributes is the summation of $U_1$ to $U_{17}$, the derivative of $\sum U_i$ ($\sum U_i'$) is calculated. The results in Figure 4 show the impact of HSR attributes on user acceptance. It shows a steadily increasing trend. When these attributes are enhanced, user acceptance increases. To improve user acceptance, they must focus their attention on the degree of these attributes when using them. Notably, when the service level exceeds 70%, acceptance accelerates, outpacing the previous increase rate. When the product attributes exceed 70%, the revenue increases more than before. To attract users and gain revenue, it is best to increase the product attributes to more than 70%.

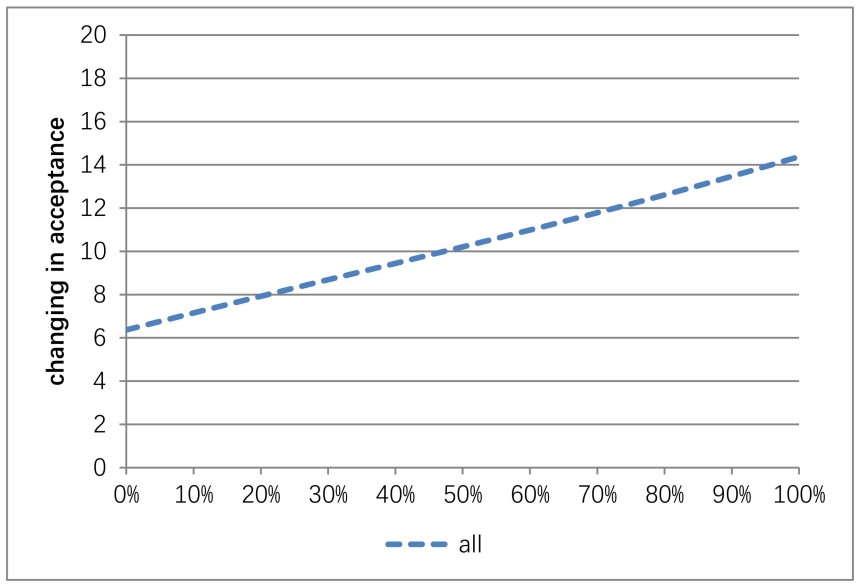

**Figure 4.** Sensitivity test of the aggregated HSR attributes.

## 5. Conclusions

As a new technology, HSR is rapidly developing worldwide. However, due to a lack of understanding of user acceptance, it is limited in its development. This issue has gradually attracted attention from HSR development companies, users, and scholars. From the perspective of perceived value, this study analyses the user's acceptance of HSRs. The various attributes that affect the user's acceptance of the HSR are analyzed, and the degree of change brought by different attributes is explored. Based on the comprehensive insights of the Kano models, this study reveals the multidimensional and perceived impact of perceived value, prompting users to embrace newer HSR technologies. Its multi-perceived value is thoroughly evaluated to determine user acceptance when HSR has differing attributes.

In the existing research, there is a lack of analysis of the impact on users' acceptance of hotel service robots from the perspective of perceived value. For studying the relationship between determinants and user acceptance of HSRs, this study uses linear and nonlinear two-dimensional quality models, addressing the shortcomings of existing studies that only study linear relationships. It analyses the HSR attributes and user acceptance in the context of the user's lack of HSR experience. In addition, it combines the quantitative insights of the Kano model through questionnaires, revealing the differing influences of HSR attributes formed in order to improve user acceptance. This study provides evidence of a nonlinear relationship between HSRs' perceived value and user acceptance.

## 6. Managerial Implications

The pattern of differences formed by acceptance has several practical implications. For developing HSRs, it is important to design optimal strategies for attribute assignment. When the existing literature explores attributes of user acceptance, most are based on traditional technology acceptance theories. Compared with the traditional linear quality model, the Kano model is a linear and nonlinear two-dimensional quality model, which can better explain the relationship between users' acceptance of HSRs and perceived value by comprehensively analyzing the perceived value of high-speed rail and the user acceptance loop, further analyzing the importance of HSR's attributes, and deeply exploring the commonly used models for analyzing user acceptance. To better develop HSRs, one must have a clear understanding of its current performance level and adjust its strategy accordingly. For HSRs, the overall performance level in the early stages of development may be relatively low. Therefore, it may be more valuable to reduce the users' unacceptability by allocating resources to the unacceptable attributes leading to lower acceptance (i.e., the attributes at the lower left of the diagonal line in Figure 2). As the technology matures and the level of HSR performance continues to improve, it may be beneficial to shift the attention to attributes that increase user acceptance (i.e., the attributes on the lower right of the diagonal line in Figure 2).

In addition, by quantifying the user's sensitivity to the performance improvement of different product attributes, some references and guidance are provided. A15 (Social interaction) was determined to form the most sensitive service attribute concerning user acceptance. Interpreting this result suggests that resources allocated to improve social interactivity yield the most effective returns (measured by user acceptance). Resource prioritization is important to maintain an optimal resource allocation plan. The results, according to Figure 3, show that hotel room price and social norms compliance should be prioritized to enhance economic and social/environment factors, respectively. Quality/function, clean, and quiet attributes are the most helpful in enhancing quality/function. Prioritization of social interaction in the emotional field can enhance EM. Therefore, when these attributes are enhanced, users will be more accepting of HSRs. Substituting these results for user acceptance of HSR attributes suggests that special efforts may be required to provide attributes that exceed user expectations.

User assessment of the acceptance of HSR attributes is a complex, multi-level, comprehensive problem that needs constant improvement and maturity. Despite this study's

contribution, there are many shortcomings in the research on the acceptance of HSR attributes. It must still face evaluation according to its limitations and needs further studies. First, although the Kano model, based on the theory of attractive mass, has been widely accepted, the quantitative method used in this study is based on several hypotheses [41]. For example, assuming the U-P function passes through the two endpoints of $(1, d_i^+)$ and $(0, d_i^-)$, the nonlinear effects of the A and M properties are expressed as an exponential function. Although these are logical assumptions, future research may lead to improvements by drawing more complex relationships.

Second, this study is only on one country (China), where the development of HSRs is still in its infancy. Therefore, regarding technological advancement and product popularity, Chinese users' product attribute requirements and perceptions may differ from those of users providing more mature services. Therefore, the specific characteristics of the Chinese context may limit the universality of the research results.

This study provides practical implications that help formulate sustainable development strategies for the hotel industry, bringing more convenience and a good sense of use to customers, HSRs will change the original consumption of staff in the hotel industry in terms of staff recruitment, staff training, assessment costs, and so forth, resulting in new operation and business methods, HSR researchers and hotel industry entrepreneurs can refer to these results to improve acceptance and operating income. To get more accurate and reliable results, more research on HSRs should be conducted. Since different robot attributes produce different effects and acceptance, different R&D improvements and marketing strategies can also be carried out concerning these results. Relevant regulatory agencies should also formulate corresponding laws and systems according to the development of the hotel industry and HSRs. With the maturity of future technology, HSRs' application to the actual environment may bring differing degrees of experiences and impacts to humans and the environment, thereby improving the quality of human life.

**Author Contributions:** Writing and data analysis, M.X.; writing and editorial review, H.-b.K. All authors have read and agreed to the published version of the manuscript.

**Funding:** This research received no external funding.

**Conflicts of Interest:** The authors declare no conflict of interest.

## Appendix A

### Questionnaire

1. Hotel room price: Amount to be paid for staying in a hotel room.
If hotel room prices for hotel service robots are not cheap, would you choose it?
If hotel room prices for hotel service robots are cheap, would you choose it?

(1). I like it that way
(2). It must be that way
(3). I am neutral
(4). I can live with it that way
(5). I dislike it that way

2. Hotel charge service price: The price you need to pay when using dry cleaning services in the hotel.
If hotel charge service prices for hotel service robots are not cheap, would you choose it?
If hotel charge service prices for hotel service robots are cheap, would you choose it?

(1). I like it that way
(2). It must be that way
(3). I am neutral
(4). I can live with it that way
(5). I dislike it that way

3. Catering price: The cost incurred in the purchase of food or drinks in the hotel.
If hotel catering prices for hotel service robots are not cheap, would you choose it?
If hotel catering prices for hotel service robots are cheap, would you choose it?

(1).   I like it that way
(2).   It must be that way
(3).   I am neutral
(4).   I can live with it that way
(5).   I dislike it that way

4. Safety: The safety of robot use.
If the hotel using hotel service robots is not safe, would you choose it?
If the hotel using hotel service robots is safe, would you choose it?

(1).   I like it that way
(2).   It must be that way
(3).   I am neutral
(4).   I can live with it that way
(5).   I dislike it that way

5. Reliability: Robots can accurately execute commands.
If the hotel uses a hotel service robot that is not reliable, would you choose it?
If the hotel uses a hotel service robot that is reliable, would you choose it?

(1).   I like it that way
(2).   It must be that way
(3).   I am neutral
(4).   I can live with it that way
(5).   I dislike it that way

6. Save time: It takes less time for the robot to execute commands.
If the hotel using the hotel service robot does not save time, would you choose it?
If the hotel using the hotel service robot saves time, would you choose it?

(1).   I like it that way
(2).   It must be that way
(3).   I am neutral
(4).   I can live with it that way
(5).   I dislike it that way

7. Personification: It feels more like a real human being.
If the hotel uses a hotel service robot that is impersonal, would you choose it?
If the hotel uses a hotel service robot that is personalized, would you choose it?

(1).   I like it that way
(2).   It must be that way
(3).   I am neutral
(4).   I can live with it that way
(5).   I dislike it that way

8. Convenience: Simpler and faster to use.If the hotel uses a hotel service robot that is an inconvenience, would you choose it?
If the hotel uses the hotel service robot that is convenient, would you choose it?
Will the convenience affect your use of hotel service robots?

9. Diversified services: More types and forms of service than ever.
If the hotel service using hotel service robots is not diversified, would you choose it?
If the hotel service using hotel service robots is diversified, would you choose it?

(1).   I like it that way
(2).   It must be that way
(3).   I am neutral
(4).   I can live with it that way
(5).   I dislike it that way

10. Cleanly: The cleanliness of the hotel environment.
If the hotel using hotel service robots is not clean, would you choose it?
If the hotel using hotel service robots is clean, would you choose it?

(1). I like it that way
(2). It must be that way
(3). I am neutral
(4). I can live with it that way
(5). I dislike it that way

11. Quietly: The surrounding environment is not noisy during the stay.
If the hotel using hotel service robots is not quiet, would you choose it?
If the hotel using hotel service robots is quiet, would you choose it?

(1). I like it that way
(2). It must be that way
(3). I am neutral
(4). I can live with it that way
(5). I dislike it that way

12. Privacy protection: The high-tech means of robots are used to make the personal privacy of customers better protected.
If the hotel using hotel service robots does not have privacy protection, would you choose it?
If the hotel using hotel service robots does have privacy protection, would you choose it?

(1). I like it that way
(2). It must be that way
(3). I am neutral
(4). I can live with it that way
(5). I dislike it that way

13. Trustworthy: It is a belief that in the uncertain environment, the user actively predicts the behavior of the robot, relies on the robot, and believes that the robot will act as expected.
If the hotel uses a hotel service robot that is not trustworthy, would you choose it?
If the hotel uses a hotel service robot that is trustworthy, would you choose it?

(1). I like it that way
(2). It must be that way
(3). I am neutral
(4). I can live with it that way
(5). I dislike it that way

14. Luxury: Magnificent, rich feeling.
If the hotel using hotel service robots is not luxurious, would you choose it?
If the hotel using hotel service robots is luxurious, would you choose it?

(1). I like it that way
(2). It must be that way
(3). I am neutral
(4). I can live with it that way
(5). I dislike it that way

15. Social interaction: Social activities that interact with other individuals for material and spiritual exchanges.
If the hotel uses a hotel service robot that cannot socially interact, would you choose it?
If the hotel uses a hotel service robot that can socially interact, would you choose it?

(1). I like it that way
(2). It must be that way
(3). I am neutral
(4). I can live with it that way
(5). I dislike it that way

16. Aesthetic/appearance: Looks comfortable and happy.
If the hotel using hotel service robots is not aesthetically pleasing, would you choose it?
If the hotel using hotel service robots is aesthetically pleasing, would you choose it?

(1).　I like it that way
(2).　It must be that way
(3).　I am neutral
(4).　I can live with it that way
(5).　I dislike it that way

17. Enjoyment: It is the feeling of pleasure and satisfaction that you have when you do or experience something that you like.
If the hotel using hotel service robots is not enjoyable, would you choose it?
If the hotel using hotel service robots is enjoyable, would you choose it?

(1).　I like it that way
(2).　It must be that way
(3).　I am neutral
(4).　I can live with it that way
(5).　I dislike it that way

18. Legal compliance: Compliance with the law.
If the hotel using hotel service robots is not in legal compliance, would you choose it?
If the hotel using hotel service robots is in legal compliance, would you choose it?

(1).　I like it that way
(2).　It must be that way
(3).　I am neutral
(4).　I can live with it that way
(5).　I dislike it that way

19. Social norm compliance: Robots can make more ethical choices (for example, if a guest falls or has a sudden illness, the robot can automatically call the police or help the guest).
If the hotel uses a hotel service robot with no social norms compliance, would you choose it?
If the hotel uses a hotel service robot with social norms compliance, would you choose it?

(1).　I like it that way
(2).　It must be that way
(3).　I am neutral
(4).　I can live with it that way
(5).　I dislike it that way

20. Reduce manufacturing waste: Precise use of computing resources without waste.
If the hotel uses a hotel service robot that does not reduce manufacturing waste, would you choose it?
If the hotel uses a hotel service robot that reduces manufacturing waste, would you choose it?

(1).　I like it that way
(2).　It must be that way
(3).　I am neutral
(4).　I can live with it that way
(5).　I dislike it that way

21. Reputation: It is famous for using the latest technology of hotel robots.
If the hotel using hotel service robots is not reputable, would you choose it?
If the hotel using hotel service robots is reputable, would you choose it?

(1).　I like it that way
(2).　It must be that way
(3).　I am neutral
(4).　I can live with it that way
(5).　I dislike it that way

22. Scientific: The use of hotel robots promotes the necessity of science and enables more people to support the use and development of new technologies.

If the hotel using hotel service robots is not scientifically motivated, would you choose it?

If the hotel using hotel service robots has scientific motivations, would you choose it?

(1).   I like it that way
(2).   It must be that way
(3).   I am neutral
(4).   I can live with it that wayC
(5).   I dislike it that way

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
