# Peer review of "User Acceptance of Hotel Service Robots Using the Quantitative Kano Model"

_sustainability, doi:10.3390/su14073988_

Round 1

Reviewer 1 Report

I found this study interesting that adds value to the field. Some contents are not clear and should be simplified in terms of writing. For instance, in lines 19 and forward, is not clear what exactly the authors are trying to communicate. Since this part of the abstract is very important to summarize the core findings of the study, rewording and reconstructing the abstract is highly recommended.

…..this study provides 19 theoretical and practical insights regarding not only the perceived value….

The same issue is with the Results section in line 363. There is a long sentence, which after several readings I failed to understand what authors were trying to communicate. Please reconstruct the content for clarity of understanding.

….A qualitative categorisation of product attributes may not adequately account for….

This article is very comprehensive that addresses all the required sections. The findings are explained in detail and the plots are very well presented. Good luck with the publishing of this manuscript!

Reviewer 2 Report

I congratulate the authors for the effort made.

The study is fascinating, different and current. I appreciate the opportunity to read and comment on the document. I congratulate the authors for the idea, and the work is done. The following suggestions could help improve the paper.

The ten main limitations of the study are, in my opinion, the following:

1) The different variables (A1, A2...) are poorly based on previous studies.
2) It would be interesting to include some conclusions and theoretical, practical, and methodological implications.
3) The chosen method is not sufficiently justified.
4) The theory of value used in the study is not sufficiently explained or justified.
5) The design of the questionnaire used in the study should be explained. What has been the process and the references used to create the items?
6) It is necessary to explain the study population and the sample design. Why this sample size and the chosen sample structure?
7) The Kano Model should be better explained.
8) References must be adapted to the requirements of the Journal.
9) The analysis carried out should be justified and explained more clearly.
10) Additional effort should be made to add clarity logical and sequential structure, from the general to the particular, linking the aspects discussed.

Reviewer 3 Report

The subject of the paper is very interesting and focuses on the implementation of new technologies, in particular artificial intelligence (AI) and robots in the hotel industry. The author analysed users acceptance of hotel service robots using the quantitative Kano model.

However, there are some shortcomings that require improvement. It is essential to complete the paper for the elements as following:

  1. Certainly, it is crucial to supplement the article with clearly formulated research hypotheses, which should then be verified on the basis of the performed statistical analyzes.
  2. It is also advisable to supplement the references with current publications. In fact, only 9 out of 34 articles from the last 5 years were referred in the publication. 

Reviewer 4 Report

Dear Authors,

First of all, congratulations for your consistent work, in an area of topicality and interest!

I find the research interesting and adds to the knowledge in the field. In my opinion the paper is well organized, brings solid arguments into discussion, and is adequately referenced. I could suggest only some minor improvements, to add more value and clarity to the paper:

  • the most important suggestions is to add the questionnaire as annex, so that readers can form a better opinion about the context of the responses analyzed
  • it would be useful to separate the conclusions of the paper in a distinct section
  • a wider discussion of the results obtained would be useful, especially in the light of similar research already published
  • it would be interesting to expand the discussion on the implications of the results for hotel industry managers and entrepreneurs, possibly also for regulators

Good luck!

Round 2

Reviewer 2 Report

We congratulate the authors for their efforts.

The document includes the changes suggested in the previous revision.

Reviewer 4 Report

Dear Authors,

Congratulations for you work and good luck with future research!

Best regards,